# Strong Negative Association between Cesarean Delivery and Early Initiation of Breastfeeding Practices among Vietnamese Mothers—A Secondary Analysis of the Viet Nam Sustainable Development Goal Indicators on Children and Women Survey

**DOI:** 10.3390/nu15214501

**Published:** 2023-10-24

**Authors:** Tam Thi Thanh Nguyen, Kimihiro Nishino, Lan Thi Huong Le, Souphalak Inthaphatha, Eiko Yamamoto

**Affiliations:** 1Department of Healthcare Administration, Nagoya University Graduate School of Medicine, 65 Tsurumai-cho, Showa-ku, Nagoya 466-8550, Japan; nguyen.thi.thanh.tam.y6@s.mail.nagoya-u.ac.jp (T.T.T.N.);; 2Department of Nutrition and Food Safety, Thai Nguyen University of Medicine and Pharmacy, Thai Nguyen 250000, Vietnam; 3Thai Nguyen General Hospital, Thai Nguyen 250000, Vietnam

**Keywords:** early initiation of breastfeeding, cesarean delivery, Vietnam, sustainable development goals, colostrum

## Abstract

Early initiation of breastfeeding (EIBF) involves feeding a newborn with the mother’s breast milk within the first hour of delivery. The prevalence of EIBF in Vietnam has recently shown a downward trend. The present study aimed to demonstrate the current prevalence of EIBF practices and identify factors associated with EIBF among Vietnamese mothers with children under 24 months of age. This study was a secondary analysis of data from the Viet Nam Sustainable Development Goal Indicators on Children and Women (SDGCW) survey 2020–2021. The study participants included 1495 mothers extracted from the SDGCW dataset. Descriptive statistics and logistic regression analyses were performed. The prevalence of EIBF practice was 25.5% among all mothers, 31.9% among vaginal-delivery mothers groups, and 9.0% among cesarean-section mothers groups. Factors negatively associated with EIBF were younger age (0.18 times), cesarean delivery (0.25 times), and absence of skin-to-skin contact with newborns immediately after birth (0.43 times). The prevalence of EIBF among Vietnamese mothers was found to be substantially low, especially among those who underwent cesarean delivery. EIBF should be promoted among younger mothers and those who underwent cesarean delivery.

## 1. Introduction

Early initiation of breastfeeding (EIBF) is defined as the provision of the mother’s breast milk to infants within the first hour of birth and ensures that the newborn receives colostrum [1]. The benefits of EIBF in improving maternal and newborn health have been well documented over the past few decades. EIBF has been reported to be noticeably effective in reducing the mortality rate of newborns with low and normal birth weights [2,3,4,5]. It is well known that colostrum is a rich source of essential nutrients and immunization factors, which boost the immune system and strengthen the gastrointestinal system of newborns [6,7]. Hence, newborns with EIBF reportedly experience a lower risk of diarrheal mortality [8,9], acute respiratory infection [10,11], and severe illness [12] in comparison to those without EIBF. Moreover, breastfeeding promotes oxytocin secretion in postpartum mothers. Therefore, mothers who practice EIBF have a lower risk of postpartum hemorrhage [13,14]. Furthermore, EIBF not only has short-term effects of increasing the interaction between mothers and newborns but also long-term effects of improving infants’ self-regulation one year after birth [15,16]. Therefore, EIBF should be highly encouraged and strongly supported for the amelioration of newborn care and maternal health [17].

Despite the abovementioned benefits of EIBF, the current global average of EIBF practices is still 51.9%, with a range of 41% to 69% [18,19]. The highest prevalence of EIBF, at around 60–80%, has been reported in a few East and South African countries as well as in some Middle Eastern Asian countries [20,21,22,23,24]. The lowest prevalence has been reported in some South American and Southeast Asian countries at approximately 30% [25,26]. Factors positively associated with EIBF practice have been reported to be a higher level of education, counseling on breastfeeding during pregnancy, non-smoking habit, absence of obstetrical complications, vaginal delivery, and early skin-to-skin contact with the newborn [27,28,29,30,31,32,33,34]. Meanwhile, mothers who belonged to minor ethnic groups, were older, delivered via cesarean section, or whose newborns had a lower Apgar score and birth weight were reported to be negatively associated with EIBF practice [27,29,31,35].

According to previous reports, a downward trend in terms of EIBF practice has been observed in Vietnam. A peak of 60.0% was reported in 2006, which reduced to 40.0% in 2011 and 26.5% in 2014. These statistics have been ranked the lowest in Southeast Asian countries [35,36,37]. Although various implications have been projected, there is scarce evidence to clarify the definitive reason for the suboptimal and downward trend in EIBF practice during the last decade or the negatively associated factors in Vietnam [35,38,39]. Therefore, to address the challenges of suboptimal EIBF practices in Vietnam, this study was undertaken to investigate the current prevalence of EIBF practices and to identify the factors associated with EIBF among Vietnamese mothers with children under two years of age.

## 2. Materials and Methods

### 2.1. Data Source

In this study, we performed a secondary analysis of the data from the survey that measured Sustainable Development Goal indicators on Children and Women (SDGCW) in Vietnam between 2020–2021. This survey was conducted by the General Statistics Office (GSO), under the Ministry of Planning and Investment, Vietnam [37]. This survey was technically and financially supported by the United Nations Children’s Fund (UNICEF) and the United Nations Population Fund (UNFPA). The Viet Nam SDGCW survey 2020–2021 was a local version of the Multiple Indicator Cluster Survey 6th Round (MICS 6) [37].

The Viet Nam SDGCW survey 2020–2021 was conducted between 18 November 2020, and 3 February 2021. The sampling method for this survey has been described in the findings report that was published in December 2022 [37]. This survey used a multi-stage, stratified cluster sampling approach to select the survey sample to obtain statistically reliable estimates of key indicators at the national level, for urban and rural areas, and for 13 analytical domains. The 13 analytical domains included 5 ethnic and 8 geographic domains [6 regions (Red River Delta, Northern Midland and Mountain areas, North Central and Central Coastal, Central Highlands, Mekong River Delta, and Southeast area) and 2 of the biggest cities (Ha Noi and Ho Chi Minh City)]. The sampling framework was based on the 2019 Vietnam Population and Housing Census [37]. The primary sampling units selected in the first stage were enumeration areas (EAs) defined for census enumeration. A specified number of EA censuses were selected systematically with probability proportional to size. Households were listed in each EA sample and a systematic sample of 20 households was selected in the second stage. A total of 700 EAs and 14,000 households were selected for interviews and data collection for this survey. Given the 13,511 occupied households, the household response rate was 98.9%, with 13,359 successfully interviewed households [37].

The questionnaires were based on the MICS6 standard questionnaires. The questionnaires from the MICS6 model English version were customized and translated into Vietnamese. Six validated questionnaires were administered to each household, including a household questionnaire, a water quality testing questionnaire, a questionnaire for individual women aged 15–49 years, a questionnaire for individual men aged 15–49 years, a questionnaire for children aged <5 years, and a questionnaire for children aged 5–17 years. Our study used the women’s dataset that was derived from women’s questionnaires. The women’s questionnaire focused on their background along with maternal and newborn health. Of the 13,359 households interviewed, 11,294 women aged 15–49 years were identified. Of these, 10,770 were individually approached and a response rate of 95.4% was obtained within the interviewed households [37].

### 2.2. Study Participants and Sample Size

From the women’s dataset of 11,294 individuals from the database of 13,359 interviewed households, we removed 9758 women who did not have any children under 24 months of age at the time of the interview. A total of 41 mothers with missing EIBF outcome values were excluded to create the final dataset. The fixed sample size for this study was 1495.

### 2.3. Variables

#### 2.3.1. Outcome Variable

The outcome variable in this study was EIBF practice. The interview of the Viet Nam SDGCW survey asked women about the timing of the first time their last child was put on their breast. In this study, mothers who practiced EIBF were defined as those who put their last child on their breasts within an hour of delivery.

#### 2.3.2. Socio-Demographic Variables of the Mothers

The following 11 socio-demographic variables of the mothers were assessed: age, ethnicity of the household head, residential region, living area, level of education, family wealth index, marital status, partner’s age, age at the time of the first delivery, parity, and intended pregnancy for the last child.

Mothers’ ages were divided into four groups: 15–17, 18–29, 30–39, and 40–49 years. The ethnicity of the household head was divided into two groups: Kinh or Chinese and others. Others included minor ethnicities such as Tay, Thai, Muong, Nung, Khmer, and Mong [40]. Residential regions were classified into four main parts based on the Viet Nam SDGCW survey: Northern Midland, Mountain, and Central Highland areas; Red River Delta and Mekong River Delta areas; North Central and Central coastal areas; and Southeast area [41]. Living areas illustrated the habitat of the mothers, rural or urban, based on the classification of the Viet Nam SDGCW survey [41]. The education level of the mother was defined as the highest grade completed and was categorized into four groups: none or pre-primary school, primary school, junior and senior high school, and higher education. Higher education referred to vocational school, college, university, and postgraduate education. Five levels of household wealth quintiles, which were classified in the Viet Nam SDGCW survey, were merged into three groups: poorest and poor, middle, and rich and richest [24]. Marital status described the relationship of the mothers with their partners. Mothers were married, living with a partner, or were not in a union. Not in a union was further categorized into single, separated, divorced, or widowed. The partner‘s age was separated by a median of 31 years. The mother’s age at the time of the first delivery was divided into three groups: <18, 18–29, and ≥30 years. Parity indicated the number of live children the mother gave birth to and was classified into three groups: one child, two to three children, and four or more children. The intended pregnancy for the last child indicated whether the last child was planned by the family (yes) or not (no).

#### 2.3.3. Antenatal Care and Delivery Variables of the Mothers

The variables that illustrated the mother’s antenatal care (ANC) for the last child included gestational week at the time of the first ANC, number of ANC visits, and ANC provider. Gestational week at the time of the first ANC of the last pregnancy was grouped into three categories: 1–12 weeks, ≥13 weeks, and none [42]. The number of ANC visits during the last pregnancy was categorized into four groups: ≥8 times, 4–7 times, 1–3 times, and none [42,43]. The ANC provider during the last pregnancy involved the person who mainly performed ANC for the mothers, namely doctors, nurses, midwives, community health workers, and none.

The delivery variables of the mothers included the place of delivery of the last child, delivery mode, and the assistant present at the time of delivery. The place of delivery of the last child could be a public hospital, another public sector, the private medical sector, a home, or other. Other public sectors included all public medical facilities other than public hospitals, such as local clinics, community health centers, hospitals of a ministry or sector, and other public institutions. Private medical sectors referred to private hospitals or other private medical institutions. Delivery modes of the last child were vaginal delivery or cesarean section. Delivery assistants for the last child were doctors, nurses and/or midwives, community health workers, and no medical support.

The other variables were skin-to-skin contact immediately after the last delivery along with the sex and birth weight of the last child. Skin-to-skin contact right after the last delivery indicated whether the child was placed on the mother’s chest immediately after the delivery (yes) or not (no). The sex of the last child was recorded as male or female. The birth weight of the last child was recorded by asking the mother about it on the interview day and was categorized into three groups: low birth weight < 2500 g, normal 2500–3999 g, and macrosomia ≥ 4000 g.

### 2.4. Statistical Analysis

The Statistical Package for Social Sciences, version 28 (IBM SPSS Inc., Armonk, NY, USA) was used for data analyses. Descriptive statistics (numbers and percentages) were calculated for all variables. Mean and standard deviations were calculated for the mother’s age and the partner’s age. Chi-square or Fisher’s exact test was performed to examine the relationship between EIBF practice and each variable. Univariate and multivariate logistic regression analyses were performed to determine the association between EIBF practice and each variable. Odds ratios (OR), adjusted odds ratios (AOR), and 95% confidence intervals (CI 95%) were calculated. All variables were entered into the logistic regression model; however, marital status was systematically removed by the software. Statistical significance was set at *p* < 0.05.

### 2.5. Ethical Considerations

Within the authority to conduct surveys, the GSO strictly complies with the provisions of the Statistics Law 2015 Clause b, Article 33 (Law No. 89/2015/QH13) on the confidentiality of information provided by the respondents [37]. The Viet Nam SDGCW survey sent a “letter to households” to explain the objectives and contents of the survey. Moreover, the letters provided the commitment of confidentiality and anonymity of information before starting the interview. Verbal consent was obtained from all the participants prior to the commencement of the study. All respondents were informed regarding voluntary participation, their right to refuse to answer all or some specific questions, and their right to stop the interview at any time. We obtained permission from the Viet Nam SDGCW survey for the use of the dataset and for the publication of this study.

## 3. Results

### 3.1. Socio-Demographic Characteristics of the Mothers

In this study, the prevalence of EIBF among 1495 Vietnamese mothers with children under 24 months of age was 25.5% (n = 381) (Table 1 and Table 2).

The mean age of the mothers who formed the study population was 28.0 ± 6.1 (Mean ± SD) years. The age of the mothers ranged between 15–47 years, and the majority of them belonged to the 18–29 years age group (57.3%). The Kinh or Chinese accounted for the main ethnic groups (55.0%). Most mothers lived in the Northern Midland, Mountain and Central Highland areas (41.3%); and rural areas (74.3%). In the context of the level of education, 51% of the mothers graduated from junior or senior high schools, while 22.2% completed primary school or had lesser education.

The poorest/poor families constituted the largest proportion of mothers (57.9%). Most mothers were married or were in a union (97.3%), delivered their first child at the age of 18–29 years (79.7%), had two or three children (58.9%), and intended to have the last child (78.9%) (Table 1).

### 3.2. ANC and Delivery Characteristics of the Mothers

Most mothers had their first ANC within 1–12 weeks (90.4%), and the majority of them received ANC more than eight times during their last pregnancy (41.7%). Most mothers were provided ANC by a doctor (86.8%), while 8% of them had no ANC visits during the last pregnancy.

In terms of the place of delivery, public hospitals and home/other accounted for 68.6% and 14.1%, respectively. More than one-fourth of the mothers underwent a cesarean section (28.1%). While most mothers were assisted by a doctor at the time of delivery (81.1%), 13.0% received no medical support for the last delivery. The percentage of mothers who experienced skin-to-skin contact immediately after their last delivery was 61.5%. The birth weight of the majority of children ranged between 2500–3999 g (91.7%) (Table 2).

### 3.3. Socio-Demographic, ANC, and Delivery Characteristics of Mothers Who Underwent Vaginal Delivery

The prevalence of EIBF among mothers with cesarean section was considerably lower (9.0%) in comparison to those with vaginal delivery (31.9%) and all mothers (25.5%), suggesting that this group contained specific factors that need to be separated. Therefore, we re-investigated the sociodemographic factors, ANC, and delivery characteristics of mothers who delivered vaginally.

The mean age of mothers who had delivered vaginally was 27.2 (±6.3 SD) years. Sociodemographic factors, ANC, and delivery characteristics of the mothers who delivered vaginally were similar to those of the all-mothers group (Appendix A). However, more than half of the mothers who delivered vaginally belonged to other minority ethnic groups (54.7%). Moreover, a higher number of mothers who delivered vaginally lived in the Northern Midland, Mountain, and Central Highland areas (47.8%), and rural areas (79.3%). The proportion of mothers who delivered vaginally and had high education was comparatively smaller (20.3%), while that of those who were poorest/poor was higher (66.0%) than that of the all-mothers group (Appendix A).

The proportion of mothers whose ANC visits were 1–3 times was higher (22.9%), while the proportion whose ANC visits were ≥ 8 times was lower (33.6%) in the vaginal-delivery mothers group compared to the all-mothers group. The proportion of mothers who delivered at home or elsewhere was larger (19.6%), and that of mothers who could not receive medical support was also larger (18.0%) in the vaginal-delivery mothers group. The vaginal-delivery group was more likely to have skin-to-skin contact after birth (67.3%) in comparison to the all-mothers group (Table 2 and Appendix A).

### 3.4. Factors Associated with EIBF among Vietnamese Women

Logistic regression analysis was performed to identify the factors associated with EIBF practice in all mothers and among those who delivered vaginally (Table 3 and Appendix A). These models were statistically significant [χ^2^ (36, N = 1495) = 153.1, *p* < 0.001, classified 76.8% of the all-mothers group; and χ^2^ (35, N = 1075) = 58.1, *p* < 0.01, classified 69.9% of the vaginal-delivery mothers group].

In this study, younger mothers (15–17 years old) were less likely to practice EIBF than older mothers in the all-mothers group [Adjusted OR (AOR) = 0.18, 95%CI 0.04–0.93] (Table 3). Mothers living in the Red River Delta and Mekong River Delta were less likely to practice EIBF in the all-mothers group (AOR = 0.18; 95%CI 0.04–0.93).

Mothers whose ANC provider was a community health worker demonstrated 8.8 times increased EIBF practice in the all-mothers group (AOR = 8.81; 95%CI 2.17–35.69) compared to those whose ANC provider was a doctor.

Mothers who delivered their last child via cesarean section demonstrated 0.25 times lower EIBF practice compared to vaginal-delivery mothers in the all-mothers group (AOR = 0.25; 95%CI 0.17–0.38). Mothers who did not have skin-to-skin contact right after delivery were significantly less likely to practice EIBF in comparison to those who had skin-to-skin contact in the all-mothers group (AOR = 0.43; 95% CI 0.28–0.64).

Factors associated with EIBF among the vaginal-delivery mothers group were identical to those among the all-mothers group (Appendix A).

## 4. Discussion

The present study demonstrated that the prevalence of EIBF among Vietnamese women with children under 24 months of age was 25.5%, 31.9%, and 9.0% among all mothers, mothers who delivered vaginally, and those who delivered via cesarean section, respectively. We found that the factors that were negatively associated with EIBF practice in Vietnamese mothers were younger age, residence in the Red River and Mekong River Delta region, the ANC provider during the last pregnancy being a doctor, cesarean section, and the absence of skin-to-skin contact with newborns immediately after birth.

The prevalence of EIBF among Vietnamese mothers was 25.5% in 2020, as demonstrated in this study, which had declined from 60.0% in 2006 and was the lowest worldwide [35,36,37]. The possible mechanisms underlying this tragic situation may be multifactorial. However, the rapid increase in the proportion of cesarean deliveries among Vietnamese mothers must have had a substantial effect. The proportion of cesarean deliveries in Vietnam drastically increased from 17% in 2011 to approximately 50% in the current time [44,45,46]. Moreover, this increase in cesarean deliveries was more evident in urban areas and at private hospitals. The reason for this rapid increase may be the preference of Vietnamese mothers in these settings due to their desire to avoid the harmful effects of vaginal delivery [44,45,46]. In the present study, 28.1% of Vietnamese mothers underwent cesarean section, and the prevalence of EIBF among mothers who underwent cesarean section was considerably lower than those who delivered vaginally (9.0% vs. 31.9%). Mothers who underwent cesarean section were, in general, more likely to not have practiced EIBF in comparison to those who delivered vaginally. This may have been because mothers and newborns are often separated after cesarean section and delay in first contact occurs due to postoperative maternal and newborn care. It is common for the prevalence of EIBF to be low in countries where the proportion of cesarean deliveries is relatively high and vice versa [47,48]. However, this is not always true. Some countries, such as Australia, New Zealand, Peru, and Chile, reportedly have a high prevalence of EIBF practice, despite a high proportion of cesarean sections, which exceeds 40% of all deliveries [49,50,51,52]. These findings suggest that the situation of EIBF in Vietnam could be improved by highlighting the importance of EIBF practice among Vietnamese mothers and healthcare workers and modifying current perinatal care in Vietnam, even under the increasing trend of the proportion of cesarean deliveries among Vietnamese mothers.

Vietnamese mothers aged less than 18 years were 0.18 times less likely to practice EIBF compared to older age groups. All mothers included in this study had given birth within 24 months; therefore, current younger mothers aged less than 18 years at the time of the interview had given birth while within school age and showed a lower prevalence of EIBF (14.3%). In mountainous and remote areas of Vietnam, getting married while within school age is a relatively common practice, especially among minor ethnicities [37]. These young mothers might be too young to acquire adequate knowledge regarding breastfeeding, including EIBF [23,30]. Addressing childbirth in younger/school-age women in Vietnam and educating teenage mothers on best childbearing practices is an urgent issue.

We found that mothers who lived in the Red River and Mekong River Delta areas were 0.18 times more likely to practice EIBF than those in other regions. This finding is consistent with that of previous reports from Vietnam. The two largest cities and the majority of private hospitals in Vietnam are located in the Red River and Mekong River Delta areas. Many mothers living in these areas prefer cesarean delivery to vaginal delivery without exact medical indications and seek private hospitals to give birth [46,53]. Indeed, in the vaginal-delivery mothers group, the decline in EIBF practice of mothers living in this area was modest, as indicated by the fact that mothers in this area who delivered vaginally were 0.87 times more likely to practice EIBF than those in other regions, indicating undetected factors other than cesarean section that need to be assessed (Appendix A).

In the context of ANC visits, women who were provided ANC during their last pregnancy by community health workers were more likely to practice EIBF in comparison to those who received ANC from doctors, nurses, or midwives (8.81 times). Community health workers are medical personnel who mainly work in the remote areas of Vietnam where cesarean sections cannot be performed. This may have positively affected EIBF practice among these mothers. Conversely, doctors, nurses, and midwives in Vietnam should convey the importance of EIBF as ANC providers in order to lead to better practices. In the present study, we did not observe any positive correlation between the number of ANC visits and the prevalence of EIBF, as reported by a few studies that were conducted in other countries [54,55].

This study had several clear advantages. First, the data were collected at the national level; hence, the findings represent all Vietnamese women with children under the age of 24 months. Second, the questionnaire used in the Viet Nam SDGCW survey was validated and customized to be appropriate for Vietnamese culture and opinions; therefore, the answers obtained had high reliability. However, this study also had some limitations which need to be considered. The cross-sectional study design could not identify the cause-and-consequence relationships among all factors. The questionnaire used in the Viet Nam SDGCW survey did not cover certain specific associated factors that may have been related to EIBF practices among Vietnamese mothers, such as knowledge of breastfeeding, breastfeeding consultation by healthcare workers, and feeding methods during the lactation period. Further studies including a wide range of factors related to EIBF practices are warranted to achieve a better understanding of the Vietnamese situation.

## 5. Conclusions

In this study, the proportion of EIBF among Vietnamese mothers with children aged <24 months was substantially low (25.5%). The prevalence of EIBF practice differed considerably depending on the delivery mode, with 31.9% in vaginal-delivery mothers and 9.0% in cesarean-delivery mothers, suggesting a downward trend in EIBF practices among Vietnamese mothers. Vietnamese mothers who were 15–17 years old, lived in the Red River and Mekong River Delta areas, received ANC by a doctor, underwent cesarean section, and did not have skin-to-skin contact with their newborns immediately after birth were less likely to practice EIBF. These findings suggest that Vietnamese mothers should be well educated about breastfeeding, and the importance of EIBF should be promoted, especially in the abovementioned settings.

## Figures and Tables

**Table 1 nutrients-15-04501-t001:** Socio-demographic characteristics of mothers who practiced early initiation of breastfeeding.

Variables		EIBF ^#^ n (%)	No EIBF ^#^ n (%)	Total N = 1495 (%)	*p*-Value
Age (years)					0.329 *
	15–17	6 (14.3)	36 (85.7)	42 (2.8)	
	18–29	217 (25.4)	639 (74.6)	856 (57.3)	
	30–39	143 (26.1)	404 (73.9)	547 (36.6)	
	40–49	15 (30.0)	35 (70.0)	50 (3.3)	
Ethnicity of household head					0.023 **
	Kinh/Chinese	190 (23.1)	632 (76.9)	822 (55.0)	
	Others ^a^	191 (28.4)	482 (71.6)	673 (45.0)	
Residential regions					0.005 *
	Northern Midland, Mountain, and Central Highlands areas	184 (29.8)	433 (70.2)	617 (41.3)	
	Red River and Mekong River Delta regions	90 (20.2)	355 (79.8)	445 (29.8)	
	North Central and Central Coastal area	51 (25.1)	152 (74.9)	203 (13.6)	
	Southeast area	56 (24.3)	174 (75.7)	230 (15.4)	
Living areas					0.734 **
	Urban	95 (24.7)	289 (75.3)	384 (25.7)	
	Rural	286 (25.7)	825 (74.3)	1111 (74.3)	
Level of education					0.284 *
	None/pre-primary	51 (30.9)	114 (69.1)	165 (11.0)	
	Primary school	46 (27.5)	121 (72.5)	167 (11.2)	
	Junior and senior high school	189 (24.8)	573 (75.2)	762 (51.0)	
	Higher education ^b^	95 (23.7)	306 (76.3)	401 (26.8)	
Family wealth index					0.127 *
	Poorest/poor	235 (27.1)	631 (72.9)	866 (57.9)	
	Middle	56 (25.8)	161 (74.2)	217 (14.5)	
	Rich/richest	90 (21.8)	322 (78.2)	412 (27.6)	
Marital status					0.357 **
	Married/in a union	368 (25.3)	1087 (74.7)	1455 (97.3)	
	Not in union ^c^	13 (32.5)	27 (67.5)	40 (2.7)	
Partner’s age ^d^ (years)					0.673 **
	<31	183 (24.8)	556 (75.2)	739 (50.8)	
	≥31	185 (25.8)	531 (74.2)	716 (49.2)	
Age at the time of the first delivery (years)					0.275 *
	<18	46 (23.1)	153 (76.9)	199 (13.3)	
	18–29	314 (26.3)	878 (73.7)	1192 (79.7)	
	≥30	21 (20.2)	83 (79.8)	104 (7.0)	
Parity					0.410 *
	1	116 (24.2)	364 (75.8)	480 (32.1)	
	2–3	225 (25.5)	656 (74.5)	881 (58.9)	
	≥4	40 (29.9)	94 (70.1)	134 (9.0)	
Intended pregnancy for the last child					0.663 **
	Yes	304 (25.8)	876 (74.2)	1180 (78.9)	
	No	77 (24.4)	238 (75.6)	315 (21.1)	
Total		381 (25.5)	1114 (74.5)	1495 (100.0)	

* Chi-square test. ** Fisher’s exact test. ^#^ Early initiation of breastfeeding. ^a^ Others included minor ethnicities such as Tay, Thai, Muong, Nung, Khmer, and Mong. ^b^ Higher education: Women who graduated from vocational school, college, or university or had a postgraduate degree. ^c^ Women were single, separated, divorced, or widowed. ^d^ Partner’s age was divided by the median.

**Table 2 nutrients-15-04501-t002:** Characteristics of antenatal care and delivery of mothers who practiced early initiation of breastfeeding.

Variables		EIBF ^#^ n (%)	No EIBF ^#^ n (%)	Total N = 1495 (%)	*p*-Value
Gestational week at the time of the first ANC ^a^ of the last pregnancy (weeks)					0.592 *
	≥13	6 (25.0)	18 (75.0)	24 (1.6)	
	1–12	340 (25.1)	1012 (74.9)	1352 (90.4)	
	None	35 (29.4)	84 (70.6)	119 (8.0)	
Number of ANC visits during the last pregnancy (times)					0.264 *
	1–3	77 (29.1)	188 (70.9)	265 (17.7)	
	4–7	122 (25.0)	366 (75.0)	488 (32.6)	
	≥8	147 (23.6)	476 (76.4)	623 (41.7)	
	None	35 (29.4)	84 (70.6)	119 (8.0)	
ANC provider during the last pregnancy					0.008 *
	Doctor	319 (24.6)	978 (75.4)	1297 (86.8)	
	Nurse/midwife	14 (25.5)	41 (74.5)	55 (3.7)	
	Community health worker	13 (54.2)	11 (45.8)	24 (1.6)	
	None	35 (29.4)	84 (70.6)	119 (8.0)	
Place of delivery of the last child					0.040 *
	Public hospital	249 (24.3)	777 (75.7)	1026 (68.6)	
	Other public sectors ^b^	48 (28.7)	119 (71.3)	167 (11.2)	
	Private medical sectors ^c^	17 (18.7)	74 (81.3)	91 (6.1)	
	Home/other	67 (31.8)	144 (68.2)	211 (14.1)	
Delivery mode of the last child					<0.001 **
	Vaginal delivery	343 (31.9)	732 (68.1)	1075 (71.9)	
	Cesarean section	38 (9.0)	382 (91.0)	420 (28.1)	
Delivery assistants for the last child					0.014 *
	Doctor	290 (23.9)	923 (76.1)	1213 (81.1)	
	Nurse/midwife only	21 (29.6)	50 (70.4)	71 (4.7)	
	Community health worker	8 (47.1)	9 (52.9)	17 (1.1)	
	No medical support	62 (32.0)	132 (68.0)	194 (13.0)	
Skin-to-skin contact immediately after the last delivery					<0.001 **
	Yes	274 (29.8)	645 (70.2)	919 (61.5)	
	No	107 (18.6)	469 (81.4)	576 (38.5)	
Sex of the last child					0.235 **
	Male	211 (26.8)	577 (73.2)	788 (52.7)	
	Female	170 (24.0)	537 (76.0)	707 (47.3)	
Birth weight of the last child (g)					0.291 *
	<2500	9 (15.8)	48 (84.2)	57 (4.4)	
	2500–3999	297 (25.0)	892 (75.0)	1189 (91.7)	
	≥4000	13 (26.0)	37 (74.0)	50 (3.9)	
Total		381 (25.5)	1114 (74.5)	1495 (100.0)	

* Chi-square test. ** Fisher’s exact test. ^#^ Early initiation of breastfeeding. ^a^ ANC: Antenatal care. ^b^ Other public sectors referred to local clinics, commune health centers, hospitals of a ministry or sector, and other public institutions. ^c^ Private medical sectors were private hospitals or other private medical institutions.

**Table 3 nutrients-15-04501-t003:** Univariate and multivariate analyses of early initiation of breastfeeding among Vietnamese mothers.

Variables		OR (CI 95%)	Adjusted-OR (CI 95%)
Age (years)			
	15–17	0.49 (0.20–1.18)	0.18 (0.04–0.93) *
	18–29	1 (reference)	1 (reference)
	30–39	1.04 (0.82–1.33)	1.09 (0.73–1.64)
	40–49	1.26 (0.68–2.36)	1.50 (0.63–3.57)
Ethnicity of household head			
	Kinh/Chinese	1 (reference)	1 (reference)
	Others ^a^	1.32 (1.04–1.66) *	1.01 (0.68–1.50)
Residential regions			
	Northern Midland, Mountain, and Central Highlands areas	1 (reference)	1 (reference)
	Red River and Mekong River Delta areas	0.60 (0.45–0.80) ***	0.18 (0.04–0.93) *
	North Central and Central Coastal areas	0.79 (0.55–1.13)	1.09 (0.73–1.64)
	Southeast areas	0.76 (0.54–1.07)	1.50 (0.63–3.57)
Living areas			
	Urban	1 (reference)	1 (reference)
	Rural	1.06 (0.81–1.38)	0.89 (0.62–1.27)
Level of education			
	None/pre-primary	1 (reference)	1 (reference)
	Primary school	0.85 (0.53–1.37)	0.72 (0.34–1.53)
	Junior and senior high school	0.74 (0.51–1.07)	0.59 (0.29–1.19)
	Higher education ^b^	0.69 (0.46–1.04)	0.66 (0.31–1.45)
Family wealth index			
	Poor/poorest	1.07 (0.76–1.50)	0.83 (0.53–1.29)
	Middle	1 (reference)	1 (reference)
	Rich/richest	0.80 (0.55–1.18)	0.83 (0.54–1.29)
Partner’s age ^c^ (years old)			
	<31	1 (reference)	1 (reference)
	≥31	1.06 (0.84–1.34)	1.28 (0.87–1.87)
Age at the time of the first delivery (years)			
	<18	0.84 (0.59–1.20)	0.73 (0.42–1.26)
	18–29	1 (reference)	1 (reference)
	≥30	0.71 (0.43–1.16)	0.60 (0.33–1.48)
Parity			
	1	1 (reference)	1 (reference)
	2–3	1.08 (0.83–1.39)	0.74 (0.52–1.06)
	≥4	1.34 (0.87–2.04)	0.70 (0.33–1.48)
Intended pregnancy for the last child			
	No	1 (reference)	1 (reference)
	Yes	1.07 (0.80–1.43)	1.01 (0.71–1.44)
Gestational week at the time of the first ANC ^d^ of the last pregnancy (weeks)			
	≥13	0.99 (0.39–2.52)	1.16 (0.39–3.49)
	1–12	1 (reference)	1 (reference)
	None	1.24 (0.82–1.87)	1.58 (0.56–4.44)
Number of ANC visits during the last pregnancy (weeks)			
	1–3	1.33 (0.96–1.83)	0.86 (0.50–1.48)
	4–7	1.08 (0.82–1.42)	0.93 (0.66–1.30)
	≥8	1 (reference)	1 (reference)
	None	1.35 (0.87–2.09)	-
ANC provider during the last pregnancy			
	Doctor	1 (reference)	1 (reference)
	Nurse/midwife	1.05 (0.56–1.95)	1.21 (0.55–2.67)
	Community health worker	3.62 (1.61–8.17) **	8.81 (2.17–35.69) **
	None	1.28 (0.84–1.93)	-
Place of delivery of the last child			
	Public hospital	1 (reference)	1 (reference)
	Other public sectors ^e^	1.26 (0.87–1.81)	1.26 (0.81–1.94)
	Private medical sectors ^g^	0.72 (0.42–1.24)	0.75 (0.42–1.36)
	Home/other	1.45 (1.05–2.01) *	0.99 (0.19–5.21)
Delivery mode for the last child			
	Vaginal delivery	1 (reference)	1 (reference)
	Cesarean section	0.21 (1.15–0.30) ***	0.25 (0.17–0.38) ***
Delivery assistant for the last child			
	Doctor	1 (reference)	1 (reference)
	Nurse/midwife only	1.34 (0.79–2.26)	0.95 (0.51–1.75)
	Community health worker	2.83 (1.08–7.40) *	1.07 (0.18–6.43)
	No medical support	1.50 (1.08–2.08) *	2.47 (0.38–15.98)
Skin-to-skin contact immediately after the last delivery			
	Yes	1 (reference)	1 (reference)
	No	0.54 (0.42–0.69) ***	0.43 (0.28–0.64) ***
Sex of the last child			
	Male	1 (reference)	1 (reference)
	Female	0.87 (0.69–1.09)	0.87 (0.66–1.15)
Birth weight of the last child (g)			
	≤2499	0.56 (0.27–1.16)	0.60 (0.27–1.30)
	2500–3999	1 (reference)	1 (reference)
	≥4000	1.06 (0.55–2.01)	1.24 (0.61–2.54)

* *p* < 0.05, ** *p* ≤ 0.01, *** *p* ≤ 0.001. ^a^ Others included minor ethnicities such as Tay, Thai, Muong, Nung, Khmer, and Mong. ^b^ Higher education: Women who graduated from senior high school, vocational school, college, university, or had a postgraduate degree. ^c^ Partner’s age was divided by median. ^d^ ANC: Antenatal care. ^e^ Other public sectors referred to local clinics, commune health centers, hospitals of a ministry or sector, and other public institutions. ^g^ Private medical sectors were private hospitals or other private medical institutions.

## Data Availability

The datasets used and/or analyzed in the current study are available from the corresponding author upon reasonable request.

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
