# Peer review of "Strong Negative Association between Cesarean Delivery and Early Initiation of Breastfeeding Practices among Vietnamese Mothers—A Secondary Analysis of the Viet Nam Sustainable Development Goal Indicators on Children and Women Survey"

_nutrients, 2023, doi:10.3390/nu15214501_

Round 1

Reviewer 1 Report

The authors investigated the underlying factors of early initiation of breastfeeding in Vietnamese mothers. It is a really interesting, well-written study.

I only have some minor comments:

1.)    It is a really hot topic and the results are really interesting. You only give maternal characteristics of all included mothers and mothers with vaginal delivery, but no such data is included for the caesarean section group. Were there any differences between the EIBF and no EIBF subgroups for this group of mothers also (you might include the data in a supplementary table)?

2.)    You also detected some factors that significantly influence EIBF. I just wonder whether these individual factors might also affect each other i.e. young mothers (under 18) with caesarean delivery and doctor as personnel in antenatal care might have a greater negative effect on EIBF than young age only? Did you measure this kind of influencing possibility with a statistic?

3.)    Discussion, line 343-348: these sentences are a bit confusing. What do you mean by recall bias? And you write mothers less than 18 years had higher prevalence of EIBF compared to teenage mothers. I don’t understand this, because both mean teenage mothers…. And which group comprised of mothers of various ages? Please clarify this section.

4.)    Discussion, line 326: there is an unnecessary letter ‘e’ after the citations [46,47], please delete it.

5.)    Conclusion, line 386: please change F to I (instead of EFBF, EIBF).

6.)    In several cases a space is missing between the text and the square brackets. Please check and correct them in the whole manuscript.

Reviewer 2 Report

Title:

I would suggest changing the Title to “Strong negative association between cesarean delivery and early initiation of breastfeeding practices among Vietnamese mothers – A secondary analysis of the Viet Nam Sustainable Development Goal Indicators on Children and Women Survey”

Abstract:

Line 23: Suggest to make two sentences. Overall practice one sentence and practice EIBF between c-section and normal delivery

Introduction

Line 53: Factors positively associated with……….? Please make it clear

Line 63: Please correct the spacing problem between countries and refs. number.

Lines 63-65: In the previous paragraph, the author mentioned associated factors of EIBF practices which included c-section delivery, in that case why this study is important. The author must mention the knowledge gaps and uniqueness of the study.

Materials and Methods

Line 89: No need to mention PSUs since it is used one time.

Line 119: The author used the same subheading in the results section (3.1.). Please use a different one. It could be the Socio-demographic variables of the mothers.

Statistical analysis

Line 176: Why was marital status automatically removed? Due to collinearity? Please clarify. Also write the estimates for instance number, %, mean, and sd as descriptive measures. Also mention the measure from simple and multiple logistic regression (e.g., OR, AOR, CI).

Results

Why the author presented Table 3 and Table 4? If the author would like to see the distribution of characteristics between c-section and vaginal delivery, then the need to present both. But why they presented the distribution only for vaginal delivery?

Again, in Table 5, the author presented the results of factors associated with EIBF among all mothers and mothers who delivered baby vaginally. Why subgroup analysis for vaginal delivery? In my understanding to answer the research objective, an analysis based on all mothers is sufficient. Also, in the results of the multiple analysis, the factors were found statistically significant among the sample of all mothers, it also found similar results for the subgroup analysis (sample of vaginal delivery). The author should clarify, otherwise, the reader will be confused. I would suggest presenting the analysis only for all samples. If the authors have any reason for presenting the subgroup analysis, they should present both (c-section vs vaginal).

Discussion

Line: Line 318: your study found the rate of c-section delivery only 28.1% but you mentioned currently c-section delivery increased to 50% (lines 312-315). What was the reason for getting a lower rate (28.1%)? You need to mention this.

Line 351: Include the reference.

The limitation section should include the recall bias issue during the collection of EIBF data.

Reviewer 3 Report

43% of the references are over 5 years old. Newer references should be used except for seminal studies.

Line 50 indicates the current global average of EIBF is 47% but the article "Global prevalence of WHO infant feeding practices in 57 LMICs in 20102018 and time trends since 2000 for 44 LMICs," published in EClinicalMedicine, 2021, states that the average is 52%, with a range of 41% to 69%.

This would be a better reference for the for the paragraph beginning on line 49.

Nguyen PH, Kim SS, Tran LM, Menon P, Frongillo EA. Early breastfeeding practices contribute to exclusive breastfeeding in Bangladesh, Vietnam and Ethiopia. Matern Child Nutr. 2020 Oct;16(4):e13012. doi: 10.1111/mcn.13012. Epub 2020 Apr 22. PMID: 32319177; PMCID: PMC7507484.

Please update your references.

Line 116 - Remove enquired and replace it with asked. remove 'put of' and replace with ".. first time their last child was put on their breast."

Line 132 and "completed" after highest grade.

I did not find other places that need editing.

Overall, this is a good study but it needs revision.

The grammar and syntax had few errors.
